# Impact of Anti-Endothelial Cell Antibodies (AECAs) in Patients with Polycythemia Vera and Thrombosis

**DOI:** 10.3390/diagnostics12051077

**Published:** 2022-04-25

**Authors:** Rossella Cacciola, Elio Gentilini Cacciola, Veronica Vecchio, Emma Cacciola

**Affiliations:** 1Hemostasis Unit, Department of Clinical and Experimental Medicine, University of Catania, 95123 Catania, Italy; 2Policlinico “Umberto I”, Department of Public Health and Infectious Diseases, “Sapienza” University of Rome, 00182 Rome, Italy; gentilini.elio@yahho.it; 3Hemostasis Unit, Medical School of Catania, University of Catania, 95123 Catania, Italy; vecchio.veronica99@gmail.com; 4Hemostasis Unit, Department of Medical, Surgical Sciences and Advanced Technologies “G.F. Ingrassia”, University of Catania, 95123 Catania, Italy; ecacciol@unict.it

**Keywords:** polycythemia vera, JAK2V617F allele burden, AECA, thrombosis

## Abstract

Polycythemia vera (PV) causes thrombosis. Erythrocytosis and cell adhesiveness are responsible for thrombosis. JAK2V617F causes inflammation and autoimmunity; however, whether or not autoimmunity or inflammation causes thrombosis has yet to be proven. In 60 PV patients, we analyzed JAK2V671F and its allele burden, autoimmune Th17 cells, interleukin-17 (IL-17), anti-endothelial cell antibodies (AECAs), endothelial leukocyte adhesion molecule-1 (ELAM-1), intercellular adhesion molecule-1 (ICAM-1), and von Willebrand factor antigen (VWF:Ag). Fifty blood donors were used as the controls. All patients were on phlebotomy-maintaining hematocrit <45% and aspirin. Of the 60 patients, 40 had thrombosis. Those patients with thrombosis had a higher JAK2V617F allele burden than those without thrombosis, andTh17 cells and IL-17 were also higher in patients with thrombosis. Interestingly, we observed a high AECA IgG ELISA ratio (ER) in patients with thrombosis, which was normal in patients without thrombosis. We found high ELAM-1 and ICAM-1 as well as high VWF:Ag in patients with thrombosis compared to patients without thrombosis. AECA-positive sera from patients with thrombosis showed enhanced binding to cytokine-treated HUVEC and a positive antibody-dependent cellular cytotoxicity, suggesting that AECA may contribute to vascular injury. A positive correlation between AECAs, allele burden, and thrombosis was found. These results suggest that autoimmunity may be an additional mechanism in PV thrombogenesis.

## 1. Introduction

Polycythemia vera (PV) is a bone-marrow malignancy, belonging to BCR/ABL-negative myeloproliferative neoplasms (MPNs), heavily burdened by thrombosis [1]. Thrombotic events may be arterial, including transient ischemic attack (TIA), stroke, myocardial infarction (MI), and peripheral arterial thrombosis, or venous, including deep vein thrombosis (DVT), superficial thrombophlebitis, and pulmonary embolism (PE), and are commonly classified as fatal or nonfatal [2]. The onset of thrombosis may occur at presentation of the disease or during follow-up [3]. However, thrombosis may also occur before diagnosis. In fact, a meta-analysis reported splanchnic vasculature thrombosis, such as hepatic or portal vein thrombosis, before a diagnosis of PV [4]. The incidence of thrombosis in patients with PV has been estimated in retrospective and prospective studies. The Gruppo Italiano Studio Policitemia Vera studied a cohort of 1213 patients with PV and found fatal and nonfatal arterial and venous thrombosis in 41% of patients, of these events 36% occurred during follow-up and 64% at presentation of the disease or before diagnosis [5]. A large study of the International Working Group for Myeloproliferative Neoplasms (IWG-MPN), including seven Italian, Austrian, and American centers, analyzed 1545 patients with WHO-defined PV. This study observed 16% arterial thrombosis and 7.4% venous thrombosis with a mortality rate of 9%. The mortality was higher in patients with previous thrombosis [6]. The ECLAP study [2] included 1638 PV patients reporting 27% arterial thrombosis and 11% venous thrombosis, in accordance with the CYTO-PV study [7]. Tefferi et al. reported a postdiagnosis thrombosis after a median follow-up of 6.9 years and treatment with cytoreductive agents, aspirin, and phlebotomy [6]. Hydroxyurea (HU) prevents arterial thrombosis, but is not effective in recurrent venous thromboembolism (VTE) or splanchnic venous thrombosis [8,9]. The efficacy of interferon-α and ruxolitinib in preventing the thrombosis is uncertain [10]. The use of low-dose aspirin (LDA), as reported in a phase III study, has weak evidence in terms of reducing the thrombotic risk and may increase the risk of hemorrhage [11]. Vitamin K-antagonists (VKAs) have been studied retrospectively, reporting an annual incidence of 5.6–6.5% VTE [9,12,13]. The association between HU and VKA [9], as well as between HU and LDA [11,14], shows little effect on preventing thrombosis and an increased risk of hemorrhage. Direct oral anticoagulants (DOACs) are currently awaiting randomized comparative trials in comparison to warfarin [14]. Phlebotomy reduces the risk of thrombosis, as shown in the CYTO-PV study [7], even if a residual risk of thrombosis in PV patients on phlebotomy has been reported [15,16]. If this is the only evidence, the prevention of thrombosis remains an unsolved clinical problem. In polycythemia vera (PV), there is inflammation [17,18], autoimmunity [19], and thrombosis [20]. A recent paper by Krečak et al. reported that PV patients with autoimmune and inflammatory disorders have an increased risk of thrombosis [21]. Inflammation is caused by the activation of the JAK-STAT3 cytokine signal pathway, responsible for an increased production of inflammatory cytokines [22]. Autoimmunity is caused by the activation of the JAK-STAT3 signaling pathway, responsible for a reduction in immunosuppressive Tregs and an increase in autoimmune Th17 cells [19,23,24]. In vasculitic diseases, such as Kawasaki disease (KD) and Henoch–Schönlein purpura, there are inflammation, autoimmunity, and thrombosis [25]. Inflammation is caused by activated helper T cells, responsible for increased production of inflammatory cytokines [26]. Autoimmunity is caused by cytokine-mediated activation of the STAT3 pathway, responsible for a reduction in immunosuppressive Tregs and an increase in autoimmune Th17 cells [27,28,29]. In these diseases, the presence of anti-endothelial cell antibodies (AECAs) and vascular thrombotic cytotoxicity has been documented [25,30,31,32]. Whether there are AECAs and vascular thrombotic cytotoxicity in PV has yet to be investigated. Therefore, we measured AECAs using an ELISA with human umbilical vein endothelial cells (HUVECs) in PV patients with thrombosis. To determine whether there was a pathogenic link between the presence of these autoantibodies and vascular damage, we examined AECA-containing sera from PV patients with thrombosis for binding of AECAs to cytokine-treated HUVECs and the antibody-dependent cellular cytotoxicity.

## 2. Materials and Methods

### 2.1. Patients

We performed a retrospective study on pre-collected samples in 60 patients (35 men and 25 women) with a PV diagnosis according to the 2016 World Health Organization (WHO) criteria [33], selected in accordance with the presence or absence of thrombosis. The inclusion period was between 1 January 2019 and 31 December 2021. The mean duration of disease was eight years (range, 2–10 years), while the median age was 55 years (range, 30–60 years). No patient had a European Leukemia Net (ELN)-determined high-risk disease [34]. No patient received cytoreductive therapy.

All patients were on phlebotomy treatments. At our institution, the phlebotomy regimen varies from 300 to 450 mL of blood withdrawn weekly or twice-weekly until the hematocrit drops below 45%; this is usually followed by maintenance phlebotomy at regular intervals every one to two months according to the hematocrit level. All patients were on low-dose aspirin (LDA). No patient had a history of thrombosis or cardiovascular risk factors, including tobacco use, hypertension, diabetes mellitus, dyslipidemia, and obesity. Nobody had leukocytosis (>11 × 10^9^/L) or TET mutation. All the patients were screened for hereditary thrombophilia, including Factor V Leiden and Prothrombin G20210 A mutation, antithrombin III, protein C and protein S deficiency, and methylenetetrahydrofolate reductase (MTHFR) mutation. Nobody had hereditary thrombophilia (Table 1). All patients underwent an internal and rheumatological visit and autoimmune and inflammatory diseases were ruled out based on clinical examinations. Fifty blood donors, age- and sex-matched, served as controls. Ethical approval for this study was obtained by the Institutional Review Board, Hemostasis and Hematology Unit, University of Catania (Q070/Q032). Each study participant provided written informed consent for study enrollment in accordance with the Declaration of Helsinki.

### 2.2. Laboratory Measurements

#### 2.2.1. JAK2V617F Mutation

JAK2V617F mutation analysis, as well as the allelic quantification, were conducted as previously described [35].

#### 2.2.2. TET2 Mutation

TET2 mutation analysis was performed by the next-generation sequencing of all the coding exons of the TET2 gene (NeoGenomics Laboratories, Fort Myers, FL, USA) [36].

#### 2.2.3. Flow Cytometric Analysis

Peripheral blood mononuclear cells (PBMCs) were freshly isolated by Ficoll density gradient centrifugation (Pharmacia, Uppsala, Sweden). The isolated PBMCs were washed twice with PBS and resuspended at 1 × 10^6^ cells/mL in complete culture medium (RPMI 1640 supplemented with 1% penicillin-streptomycin, 2 mM L-glutamine, and 10% heat-inactivated fetal bovine serum; Gibco BRL, Gaithersburg, MD, USA). For Th17 cells analysis, IL-17-producing CD3+ lymphocytes were detected as previously described [37]. PBMCs were stimulated for 4 h with a 2 μ/mL leukocyte activation cocktail (BD Pharmingen™, San Diego, CA, USA) at 37 °C and 5% CO_2_. Upon harvesting, the cells were then washed twice using PBS and surface-stained with APC-conjugated anti-CD8 (BD Pharmingen™, USA) and PerCP-conjugated anti-CD3 (BD Pharmingen™, USA) for 20 min at room temperature in the dark. Following surface staining, the cells were incubated with PE-conjugated anti-IL-17A (BD Pharmingen™, USA) after fixation and permeabilization using IntraPrep Permeabilization Reagent (Beckman Coulter Inc., Paris, France).

#### 2.2.4. Interleukin-17 (IL-17)

The serum levels of IL-17 were determined by enzyme-linked immunosorbent assay (ELISA), following the manufacturer’s instructions (Biosource, Nivelles, Belgium).

#### 2.2.5. Anti-Endothelial Cell Antibodies (AECAs)

##### Cell Culture

Human umbilical vein endothelial cells (HUVECs) were isolated as previously described [38] and cultured under standard conditions [39]. The cells were used between passages two and four. For the ELISA and cytotoxicity studies, the cells were used at confluence on gelatin-coated 96-well microtiter plates [30]. For the adsorption studies, the cells were grown to confluence on microcarrier beads (Biosilon; Nunc) as previously described [40].

#### 2.2.6. ELISA for AECAs

The technique used was described in detail for IgG [31]. Briefly, serum samples were diluted in phosphatase-buffered saline (PBS) containing 1.0% bovine serum albumin (BSA) (fraction V; Sigma) and 5 mM EDTA, at 1/1000 for IgG detection, and incubated for 60 min at room temperature with glutaraldehyde-fixed monolayers of HUVECs. After washing and drying, bound immunoglobulins were detected by a further 60 min incubation period with peroxidase-coupled rabbit anti-human IgG (Dako), and subsequent quantification of peroxidase using OPD and H_2_O_2_. All assays included at least one positive and one negative control sample of each plate. The results are expressed as an ELISA ratio (ER) calculated as ER = 100 × (S − A)/(B − A), where S is the absorbance of the sample, and A and B are the absorbances of the standard negative and positive controls. The samples were recorded as positive if the ER was greater than the mean + 3 SD of the normal group.

#### 2.2.7. Adsorption Studies

IgG prepared from AECA-positive sera by sodium sulfate precipitation was extensively dialyzed against PBS. Two milliliter samples were recirculated for 60 min at 0.25 mL/min through a small (8.8 mL) column packed with micro-carrier beads bearing glutaraldehyde-fixed confluent monolayers of HUVECs. Each sample was sequentially exposed to three such columns, and tested for depletion of anti-HUVEC IgG. The adsorbed antibodies were eluted, after four washes of the column with diluting buffer, into 500 μL of 0.1 M glycine-HCL buffer (pH 3.0), and the pH was then adjusted to 7.0 and concentrated back to the original protein concentration. In all instances, the eluate retained AECA activity when tested by ELISA.

#### 2.2.8. TNF and IL-1 Treatment of HUVECs

The HUVECs were pre-activated with the recombinant human cytokines TNF or IL-1. TNF-α (Genzyme) (specific activity 2 × 10^7^ U/mg) was used at 50 U/mL to activate the HUVECs for 4 h. IL-1α (Roche) (specific activity 3 × 10^8^ U/mg) was used at 100 U/mL for 4 h. Following activation, the HUVECs were fixed for use in the ELISA.

#### 2.2.9. Antibody-Dependent Cellular Cytotoxicity

The HUVECs were labeled with ^111^In-oxine as previously described [32], followed by incubation for 30 min with immunoglobulin or serum samples diluted in RPMI/4% BSA at the indicated concentrations. Then, 2.5 × 10^5^ cells per well were added (in 200 μL of RPMI/5% heat-inactivated fetal bovine serum for peripheral blood mononuclear cells and 200 μL of RPMI/4% BSA for neutrophils). The plates were centrifuged for 5 min at 200× *g*, incubated for 4 h at 37 °C and 5% CO_2_, then centrifuged a second time. Following this, 100 μL of each supernatant were aspirated and counted in a gamma counter. A percent lysis index was determined by calculating 100 × (experimental release—spontaneous release/maximum release—spontaneous release) [41,42].

#### 2.2.10. Adhesion Molecules

Enzyme-linked immunosorbent assay (ELISA) was used to determine the serum levels of ELAM-1 (Diaclone, France) and ICAM-1 (Thermo Fisher Scientific Inc., Waltham, MA, USA).

#### 2.2.11. Von Willebrand Factor Antigen (VWF:Ag)

Enzyme-linked immunosorbent assay (ELISA) was used to determine the serum levels of VWF:Ag (Thermo Fisher Scientific Inc., Waltham, MA, USA).

## 3. Results

In 60 patients, after a median follow-up of 6.5 years, 40 out of 60 (67%) (25 men and 15 women; mean age 52 years (range, 40–55 years)) had thrombosis. Of these, 30 had arterial thrombosis, including 10 transient ischemic attack (TIA), 5 stroke, and 15 myocardial infarction (MI), while 10 had venous thrombosis including deep vein thrombosis (DVT) (Table 2).

Additionally, 20 out of 60 (33%) (12 men and 8 women; mean age 50 years (range, 35–52 years) did not develop thrombosis. Computed tomography (CT) was used for TIA and stroke diagnosis and confirmed by magnetic resonance imaging (MRI). Electrocardiogram and duplex ultrasound were used for diagnosis of MI and DVT, respectively.

A higher JAK2VF allele burden was observed in patients with thrombosis than in patients without thrombosis (75% vs. 25%), in association with the higher count of Th17 cells in patients with thrombosis (0.85 ± 0.48%) compared to patients without thrombosis (0.40 ± 0.22%) and the controls (0.27 ± 0.20). The IL-17 levels were significantly different between patients with thrombosis (6.30 ± 0.8 pg/mL), without thrombosis (3.20 ± 0.9 pg/mL), and the controls (2.10 ± 0.3 pg/mL) (Table 3).

### 3.1. Presence of Anti-HUVEC Antibodies

The IgG ER values (mean ± *s.d.*) from the sera of the controls (*n*. 50) were 1.7 ± 2.2. Controls were AECA-negative. Samples were considered positive if the titers were greater than the mean + 3 *s.d*. The IgG ER values from the sera of the patients with thrombosis (*n*. 40) was 25.1 ± 10.1. The patients with thrombosis were AECA-positive. The IgG ER values from the sera of the patients without thrombosis (*n*. 20) was 1.8 ± 2.0. The patients without thrombosis were AECA-negative. These results are summarized in Table 4. Although dilutions of 1/1000 for IgG were routinely used, positive binding could be detected using dilutions of >1/16,000 (data not shown).

### 3.2. Binding of Anti-HUVEC Antibodies

In additional experiments, the effect of cytokine activation of HUVECs on the binding of both AECA-positive and AECA-negative samples was determined. Within each experiment, cytokine-treated or untreated cells were derived from the same source. Twenty AECA-negative serum samples from those patients without thrombosis that showed no significant binding to untreated HUVECs also failed to show binding to HUVECs that had been pretreated with TNF or IL-1. The ER values for binding to untreated HUVECs was 5.3 ± 4.5, to TNF-treated HUVECs was 4.8 ± 5.0, and to IL-1 treated HUVECs was 5.2 ± 4.3. All 40 samples from those patients with thrombosis that were positive for IgG AECAs showed increased binding when the HUVECs were pretreated with these cytokines compared to untreated HUVECs (Table 5). A mouse monoclonal antibody to MHC class I antigens (W6/32, (Serotec) was used to block the binding of AECAs to HUVECs.

### 3.3. Cytotoxicity Studies

The serum samples of the patients with thrombosis that showed positive binding for IgG AECAs were tested for antibody-dependent cellular cytotoxicity against HUVECs using peripheral blood mononuclear cells. All samples showed a lysis index greater than 15%.

### 3.4. ELAM-1 and ICAM-1 Levels

The levels of ELAM-1 were highly elevated in patients with thrombosis (60 ± 5 ng/mL) compared to those without thrombosis (30 ± 1 ng/mL) and the controls (25 ± 5 ng/mL) (*p* < 0.0001). In those patients with thrombosis, the levels of ICAM-1 were highly elevated (180 ± 10 ng/mL) compared to those without thrombosis (90 ± 10 ng/mL) and the controls (80 ± 10 ng/mL) (Table 3).

### 3.5. VWF:Ag Levels

The VWF:Ag levels were significantly elevated in the patients with thrombosis (75 ± 10 ng/mL) and normal in those without thrombosis (25 ± 2 ng/mL) and the controls (18 ± 2 ng/mL) (Table 3).

### 3.6. Statistical Analysis

The results are provided as the mean ± standard deviation, analyzed using Student’s *t*-test, correlation coefficients using Pearson’s test for parametric distributions or Spearman’s test for nonparametric distributions, and Fisher’s exact test. A *p*-value of <0.05 was considered statistically significant. The data were analyzed using SPSS 21.0 for Windows (SPSS Inc., Chicago, IL, USA).

### 3.7. Correlation Studies

A positive correlation was found between the JAK2V617F allele burden and Th17 and between Th17 and IL-17, as well as between AECAs and the allele burden and between AECAs and thrombosis. A positive correlation was identified between ELAM-1 and ICAM-1 and VWF and between VWF and the allele burden and thrombosis (Table 5).

## 4. Discussion

The traditional Virchow’s triad, including abnormal blood composition, abnormal blood flow, and abnormal vessel wall, is a useful model in the study of the mechanisms underlying thrombogenesis [43]. In PV, thrombogenesis is linked to JAK2V617F mutation-dependent clonal myelopoiesis, characterized by erythrocytosis and blood cells activation (abnormal blood composition), blood flow turbulence (abnormal blood flow), and vessel wall adhesivity (abnormal vessel wall) [44]. In particular, elevated hematocrit causes hyperviscosity and high blood flow resistance (“high flow shear”), as well as adhesive collisions between platelets, leukocytes, and the vessel wall (“high wall shear”) [7,45]. In addition, JAK2V617F mutation phosphorylates the erythroid adhesion receptor Lutheran/basal cell adhesion molecule (Lu/BCAM) of the endothelial laminin, resulting in an adhesive interaction between red cells and endothelial cells [46]. Thrombogenesis may also include inflammation and autoimmunity. In inflammatory diseases, such as systemic lupus erythematosus (SLE), Crohn’s disease, and rheumatoid arthritis (RA), interleukin-6 (IL-6), as well as IL-1β, IL-21, IL-23, and TNF-α, phosphorylates and activates STAT-3 (signal transducer and activator of transcription 3), inhibiting differentiation in immunosuppressive T regulatory (Treg) cells and stimulating differentiation in autoimmune Th17-cells [23]. Cytotoxic anti-endothelial cell antibodies (AECAs) have been reported in these diseases in association with an increased expression of adhesion molecules, such as vascular cell adhesion molecule-1 (VCAM-1), intercellular adhesion molecule-1 (ICAM-1), and E-selectin, and vascular injury [47]. In similarity to inflammatory diseases, in PV, JAK2V617F-mediated inflammation activates STAT-3 signal transduction, reducing immunosuppressive Treg cells and increasing autoimmune Th17 cells [19]. Whether there are AECAs and endothelial injury in PV remains to be investigated. First, we studied the IL-17/Th17 axis. Keohane et al. [19] studied the IL-17/Th17 axis for its tumor-immune surveillance role in patients with PV with JAK inhibitor (JAKi) and without JAKi. They found lower levels of IL-17 and higher Th17 cells in patients with JAKi than in patients without JAKi. We studied the IL-17/Th17 axis for its thrombosis-immune surveillance role in PV patients with and without thrombosis. We found higher levels of IL-17 and Th17 cells in PV patients with thrombosis than in patients without thrombosis. Second, we investigated AECAs. Hasselbalch et al. [48] found immune complexes (ICs) in MPNs, reflecting a circulating inflammatory immune state most severe in more progressive diseases. We found AECAs reflecting a circulating inflammatory immune state most severe in PV complicated by thrombosis. Barcellini et al. [49] reported autoimmunity and inflammation characterized by the presence of organ- or non-organ-specific autoantibodies and elevated IL-17 as important pathogenetic factors in MPNs. We also found autoimmunity and inflammation characterized by AECAs and elevated IL-17 as important pathogenetic factors in PV with thrombosis. Masciulli et al. [10] published a systematic review and meta-analysis on ruxolitinib (Ruxo) versus the best available therapy (BAT) in patients with PV and thrombosis. The authors analyzed the records of 663 PV patients from four randomized controlled trials with, as an end point, the molecular response (MR) and thrombosis, and found that Ruxo was more effective than BAT in terms of MR and thrombosis. Our study analyzed the records of 60 patients with PV with, as an end point, the allele burden and thrombosis. We found that an allele burden >50% was more thrombotic than an allele burden <50%, and we also observed a positive correlation between AECAs and the allele burden and thrombosis. Bjorn et al. [50], Andersen et al. [51], and Barbui et al. [52] reported that in MPNs, elevated levels of YKL-40 and C-reactive protein (CRP), respectively, might be markers of chronic inflammation and atherosclerosis. Interestingly, they found a correlation between YKL-40 and CRP and the JAK2V617F allele burden and thrombosis. We reported elevated levels of an acute-phase reactant and an indicator of vascular injury, such as VWF, that might be a marker of chronic inflammation and atherosclerosis in PV patients with thrombosis. Interestingly, we found a correlation between VWF and the JAK2V617F allele burden and thrombosis. Additionally, Krečak et al. reported an association between YKL-40 and inflammation and cardiovascular risk [53]. We reported an association between VWF and inflammation and cardiovascular risk with the occurrence of MI in 15 out of 30 of the studied PV patients. CRP was also associated with shortened leukemia-free survival in myelofibrosis. We also observed VWF associated with shortened thrombosis-free survival in AECA-positive PV patients. A large Danish epidemiological study [54] reported that JAK2V617F mutation increased the cardiovascular risk. We reported that JAK2V617F mutation increases AECAs and cardiovascular risk. In fact, in our study group, 15 out of 30 of the arterial thrombosis occurrences were MI-positive. DaSilva et al. [55] reported that patients with PV had a TNF-α-mediated endothelial activation marked by the expression of adhesion molecules such as VWF, VCAM-1, and P-selectin. We reported that patients with thrombosis have AECA-mediated endothelial activation marked by the expression of adhesion molecules such as ELAM-1, ICAM-1, and VWF. A large Swedish epidemiologic study [56] reported that chronic immune stimulation may develop myeloid malignancies. We reported that JAK2V617F-mediated chronic immune stimulation may develop AECA-related autoimmunity in PV. It is known that AECAs are strictly linked to vasculitis. Some studies have attempted to identify the endothelial antigenic targets recognized by AECAs in the sera from patients with SLE [30]. These studies showed that AECAs recognize different antigens localized in the cytoplasm of endothelial cells [30]. We attempted to identify the endothelial antigenic target recognized by AECAs. In our study, we observed an enhanced binding of AECAs to cytokine-treated HUVECs in the sera from PV patients with thrombosis. This finding may suggest that the AECAs present in PV patients with thrombosis recognize the endothelial cells activated by cytokines and that AECA-mediated endothelial alterations may occur in PV thrombosis, as evidenced by a high VWF. Further studies are to be encouraged to identify the antigenic targets expressed by cytokine-activated endothelial cells. The present study indicates that AECAs may be a manifestation of the vascular disease PV, as supported by positive antibody-dependent cellular cytotoxicity.

## 5. Conclusions

In the present study, we reported that AECAs exist in the sera of patients with PV and that AECAs are linked to thrombosis. This clinical observation may support the concept that autoimmunity and inflammation may be important factors in the pathogenesis of thrombosis in PV. Therefore, we think that the measurement of AECAs in PV patients could be recommended, particularly in those that still have thrombosis following cytoreductive and antiplatelet treatment. Statins are potent anti-inflammatory agents that also show a synergistic effect with JAK inhibition. Therefore, a role of statins in association with JAK2/1 inhibitors could be envisaged in future PV treatment.

The limitations of this study are that no causal relationship could be proven due to the study design and the small number of patients; thus, validation on larger datasets is needed. Nevertheless, these carefully collected data from a group of PV patients presenting thrombosis could represent a subgroup of PV patients with particularly high risk of thrombosis for whom a closer follow-up might be warranted.

## Figures and Tables

**Table 1 diagnostics-12-01077-t001:** Characteristics of the 60 patients with polycythemia vera and the controls.

	Pts with Thrombosis	Pts without Thrombosis	Controls
	40/60	20/60	50
**Age, y**	52 (40–55)	50 (35–52)	51 (32–52)
Men, *n*	25	12	20
Women, *n*	15	8	20
JAK2V617F allele burden	75	25	
** Risk factors **			
Tobacco use	Absent	Absent	Absent
Hypertension	Absent	Absent	Absent
Diabetes	Absent	Absent	Absent
Dyslipidemia	Absent	Absent	Absent
Obesity	Absent	Absent	Absent
Hereditary thrombophilia	Absent	Absent	Absent
Leukocytosis (>11 × 10^9^/L)	Absent	Absent	Absent
TET mutation	Absent	Absent	
** Comorbidities **			
Autoimmune disease	Absent	Absent	Absent
Kawasaki disease	Absent	Absent	Absent
Henoch–Schönlein	Absent	Absent	Absent
Systemic lupus erythematosus	Absent	Absent	Absent
Crohn’s disease	Absent	Absent	Absent
Rheumatoid arthritis	Absent	Absent	Absent
** Therapy **			
Phlebotomy	40/60	20/60	
Aspirin	40/60	20/60	

**Table 2 diagnostics-12-01077-t002:** Thrombotic events in 40 out of 60 PV patients.

Patients		
	* **n** *	**%**
**Arterial thrombosis**	30/40	75
Transient ischemic attack	10	33
Stroke	5	16
Myocardial infarction	15	50
** Venous thrombosis **	10/40	25
Deep vein thrombosis	10	25

**Table 3 diagnostics-12-01077-t003:** Laboratory characteristics and statistics of the 60 PV patients.

	With Thrombosis	Without Thrombosis	*p*
	(*n* = 40)	(*n* = 20)	
JAK2 allele burden (%)	75	25	<0.001
Th17 (%)	0.85 ± 0.48	0.40 ± 0.22	<0.004
Allele burden/Th17	Correlation	Correlation	<0.003
IL-17 (pg/mL)	6.30 ± 0.8	3.20 ± 0.9	<0.005
Th17/IL-17	Correlation	Correlation	<0.001
AECA (ER)	25.1 ± 10.1	1.8 ± 2.0	<0.005
AECA/allele burden	Correlation	Correlation	<0.0003
AECA/thrombosis	Correlation	Correlation	<0.0015
ELAM-1 (ng/mL)	60 ± 5	30 ± 1	<0.0001
ICAM-1 (ng/mL)	180 ± 10	90 ± 10	<0.0001
VWF (ng/mL)	75 ± 10	25 ± 2	<0.0001
ELAM-1/ICAM-1/VWF	Correlation	Correlation	<0.001
VWF/allele burden	Correlation	Correlation	<0.003
VWF/thrombosis	Correlation	Correlation	<0.002

**Table 4 diagnostics-12-01077-t004:** Anti-endothelial cell antibodies (AECAs) in the 60 PV patients and controls.

	ELISA Ratio
Antibody class	IgG
Controls *n*. 50	1.7 ± 2.2
Pts with thrombosis *n*. 40/60	25.1 ± 10.1
Pts without thrombosis *n*. 20/60	1.8 ± 2.0

**Table 5 diagnostics-12-01077-t005:** Enhanced binding of anti-endothelial cell antibodies (AECAs) to cytokine-treated HUVECs in AECA-positive samples from patients with thrombosis.

IgG ELISA Ratio		
Treatment	Median	Range
TNF-α (50 U/mL)	39	16–80
IL-1 (100 U/mL)	37	13–79
Untreated	15	0–20

## Data Availability

The data are not publicly available due to privacy.

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
