# Peer review of "Impact of Anti-Endothelial Cell Antibodies (AECAs) in Patients with Polycythemia Vera and Thrombosis"

_diagnostics, 2022, doi:10.3390/diagnostics12051077_

Round 1

Reviewer 1 Report

I have read this small study with clinically important findings with great interest. The link between thrombosis and autoimmunity in PV is being increasingly recognized. I find the paper interesting but suggest following improvements:

1) introduction in written in somewhat absolute statements. I suggest this to be ameliorated. Actually association of reduced thrombotic risk with phelobotomies have been shown in CYTO-PV study 

2) authors ommited the recently published clinically important paper (Krecak I et al. Autoimmune disorders and the risk of thrombotic events in polycythaemia vera. Leuk Res. 2021;110:106667. doi: 10.1016/j.leukres.2021.106667.) that reported how PV patients with autoimmune disorders experience increased thrombotic risk. Also, association of YKL-40 with inflammation and cardiovascular risk was shown by larger number of author groups (Krecak I et al. Circulating YKL-40 in Philadelphia-negative myeloproliferative neoplasms. Acta Clin Belg. 2021;76(1):32-39. doi: 10.1080/17843286.2019.1659467.) I suggest including these observations in introduction and discussion sections.

3) paper is missing several key information points: how many patients received cytoreductive therapies, how many patients had ELN determined high risk disease. There also seems to by typing error in table 1 where high number of numbers is missing and only 0 is written. In suggest to include P values for differences between groups as an additional column. Also a number of information is missing from Table 3. 

4) just curiosity, how come CT has not been used for TIA/stroke diagnosis in none of the patients.

5) A portion of text describing patients' characteristics should be moved from methods to results. Also statistical methods are given at the end of results. This should be corrected and methods and results reorganized. 

6) Disscusion is missing limitations section. It should be clearly stated that no causal relationship can be proven from the study design. Paper also does not mention, but it seems that the study was retrospectivelly performed on precollected samples. Thrombosis is evaluted as a binary event, not prior to or after diagnosis or the special referral time point, ignoring censored data due to uneven follow-up. Sample size is small with insufficient statistical power for some of the analyses. 

Author Response

Revised Manuscript Diagnostics-1676439

Article“Impact of the Anti-Endothelial Cell Antibodies (AECAs) in Patients with Polycythemia Vera and Thrombosis

point-by-point summary of responses

Author’s Reply to the Review Report (Reviewer 1)

1)Introduction in written in some what absolute statements. I suggest this to be ameliorated. Actually association of reduced thrombotic risk with phlebotomies have been shown in CYTO-PV study.

P2 – L61

The sentence “….does not low the thrombotic risk….”was deleted and written as follows:

“….has weak evidence in terms of reducing the thrombotic risk”

P2 – L66-67

The sentencePhlebotomy reduces the red cell mass but it does not reduce the risk of thrombosis” was deleted and written as follows:

“Phlebotomy reduces the risk of thrombosis, as shown in the CYTO-PV study [7], even if a residual risk of thrombosis in PV patients on phlebotomy has been reported”.

2)Authors ommited the recently published clinically important paper(Krecak I et al. Autoimmune disorders and the risk of thrombotic events in polycythaemia vera. Leul Res. 2021; 110;106667.doi: 10.1016/j.leukres.ì2021.106667.). that reported how PV patients with autoimmune disorders experience increased thrombotic risk. Also, association of YKL-40 with inflammation and cardiovascular risk was shown by larger number of author groups (Krecak I et al. Circulating YKL-40 in Philadelphia-negative myeloproliferative neoplasms. Acta ClinBelg. 2021; 76(1):32-39. Doi:10.1080/17843286.2019.1659467.) I suggest including these observations in introduction and discussion section.

P2 – L70

The sentence was added as follows:

“A recent paper by Krečak et al. reported that PV patients with autoimmune and inflammatory disorders have an increased risk of thrombosis [21]”

The reference [21] “Krečak I, Morić-Perić M, Coha B, et al. Autoimmune disorders and the risk of thrombotic events in policythaemia vera. Leuk Res 2021; 110:1-3” was quoted in the list of references.

P8 – L368

The sentences were added as follows:

“Additionally, Krečak et al. reported an association between YKL-40 and

inflammation and cardiovascular risk [53]. We reported an association between

VWF and inflammation and cardiovascular risk with the occurrence of MI in

15/30 of the studied PV patients.

Also, Krečak et al reported an association

of YKL-40 with inflammation and cardiovascular risk” [51].

“We report an association of VWF with inflammation and cardiovascular risk occurring MI in 15/30 of studied PV patients”.

The reference [53] “Krečak I, Gverić-Krečak V, Lapić I, et al. Circulating YKL-40 in Philadelphia-negative myeloproliferative neoplasms. ActaClinBelg 2021; 76(1):32-39” was quoted in the list of references.

P2 –L91

3)Paper is missing several key information points: how many patients received cytoreductive therapies, how many patients had ELN determined high risk disease. There also seemsto by typing error in table 1 where high number of numbers is missing and only 0 is written. In suggest to include P values for differences between groups as an additional column. Also a number of information is missing from Table3.

These sentences were added as follows:

No patient had European Leukemia Net (ELN) disease determined to be high risk [34].

The reference [34]“Barbui T, Barosi G, Birgegard G, et al. Philadelphia-Negative

Classical Myeloproliferative Neoplasms: Crotical Concepts and Management

Recommendations from European Leukemia Net. J ClinOncol 2011; 29:761-

770”was quoted in the list of references.

“No patient received cytoreductive therapy”.

In Table 1 the number “0” was replaced with the word “Absent”.

P values for differences between groups are in the column of the Table 3.

In Table 3 the missing information was written with the word “Correlation”.

4)Just curiosity, how come CT has not been used for TIA/stroke diagnosis in

none of the patients.

P3 – L134

The sentence“Magnetic Resonance Imaging (MRI), electrocardiogram and duplex ultrasound were used to diagnosis TIA, stroke, MI and DVT, respectively” was deleted and written as follows:

“Computed tomography (CT) was used for TIA and stroke diagnosis and confirmed by magnetic resonance imaging (MRI). Electrocardiogram and duplex ultrasound were used for diagnosis of MI and DVT, respectively”.

P2 – L96

5)A portion of text describing patients’ characteristics should be moved from methods to results. Also statistical methods are given at the end of results. This should be corrected and methods and results reorganized.

The period“In 60 patients, after a median follow-up of 6.5 years, 40/60 (67%) (25 men, 15 women; mean age 52 years (range 40-55) had thrombosis. Of these 30 had arterial thrombosis including 10 transient ischemic attack (TIA), 5 stroke, and 15 myocardial infarction (MI) and 10 had venous thrombosis including deep vein thrombosis (DVT) (Table 2). 20/60 (33%) (12 men, 8 women; mean age 50 years (range 35-52) did not develop thrombosis. Computed Tomography (CT) was used for TIA and stroke diagnosis and confirmed by Magnetic Resonance Imaging (MRI). Electrocardiogram and duplex ultrasound were used to diagnosis of MI and DVT, respectively” has been moved from P3 – L129-135 “methods” to P5 – L223 “results”.

P7 – L309

Statistical analysis has been moved at the end of results.

6)Discussion is missing limitations section. It should be clearly stated that no causal relationship can be proven from the study design. Paper also does not mention, but it seems that the study was restrospectively performer on precollected samples. Thrombosis is evaluated as a binary event, not prior to or after diagnosis or the special referral time point, ignoring censored data due to unevent follow-up. Sample size is small with insufficient statistical power for some of the analyses.

P9 –L394

Limitations section was added and written as follows:

“The limitations of this study are that no causal relationship couldbe proven due to the study design and thesmall number of patients; thus, validation on larger datasets is needed. Nevertheless, these carefully collected data from a group PV patients presenting thrombosis could represent a subgroup of PV patients at particularly high risk of thrombosis for whom a closer follow-up might be warranted”.

P2 –L89

In Materials and Methods” section P2-L89 is mentioned the study design as follows:

“We performed a retrospective study on precollected samples…..”

Reviewer 2 Report

Dear authors,

After reading the article my comments are:

  • the introduction contains too many numbers, percentages which are difficult to follow. I suggest that only the most relevant ones should be mentioned.
  • it is not clear what type of study you performed, either prospective or retrospective. How did you select the patients? Which was the inclusion period?
  • it is mentioned that none of the patients had hereditary thrombophilia. What type of thrombophilia did you test? All of them were screened for hereditary thrombophilia?
  • Autoimmune and inflammatory diseases were ruled out based on clinical examinations?
  • I also suggest that the article should be checked by a native English speaker as there are too many grammar and spelling errors.
  • The authors' contribution does not state who actually performed the study. Moreover I do not see the justification for which all authors are main authors. This may be considered a form of scientific misconduct.

Author Response

Revised Manuscript Diagnostics-1676439

Article “Impact of the Anti-Endothelial Cell Antibodies (AECAs) in Patients with Polycythemia Vera and Thrombosis

point-by-point summary of responses

Author’s Reply to the Review Report (Reviewer 2)

1)The introduction contains too many numbers, percentages which are difficult to follow. I suggest that only the most relevant ones should be mentioned.

P1 –L45

The sentence“….in 19% of patients….” was deleted.

P1-2 – L46-48

The sentence “This study also reported that 30% of 145 arterial thrombosis and 11% of 87 venous thrombosis were fatal, representing 24% and 6% of death, respectively” was deleted.

P2 – L50

The sentence “This study observed arterial thrombosis in 246 patients (16%) and venous thrombosis in 114 patients (7.4%) and that the thrombosis caused the death in 9% of patients” was written without numbers as follows:

“This study observed 16% arterial thrombosis and 7.4% venous thrombosis with mortality of 9%”.

P2 – L55

The sentence “….that showed 17% of arterial thrombosis and 12% of venous thrombosis” was deleted.

P2 – L56

The sentence…..annual rate of 2.62% ..... “was deleted.

2)It is not clear what type of study you performed, either prospective or retrospective. How did you select the patients? Which was the inclusion period?

The type of study was retrospective and this is mentioned in “Materials and Methods” section P2 – L89 as follows:

“We performed a retrospective study on precollected samples…..”

The selection of the patients is mentioned in“Materials and Methods” section P2 -L90 and written as follows:

“….selected in accordance with the presence or absence of thrombosis”.

The period of inclusion is mentioned in“Materials and Methods” section P2 -L91 and written as follows:

“The inclusion period was 1 January, 2019 and 31 Decembre, 2021”.

3)It is mentioned that none of the patients had hereditary thrombophilia. What type of thrombophilia did you test? All of them were screened for hereditary thrombophilia?

In “Materials and Methods” section P3–L138 the type of thrombophilia and screened patients was written as follows:

“All the patients were screened for hereditary thrombophilia, including Factor V Leiden and Prothrombin G20210 A mutation, antithrombin III, protein C and protein S deficiency, and methylenetetrahydrofolate reducatase (MTHFR) mutation”.

4)Autoimmune and inflammatory diseases were ruled out based on clinical examinations?

In “Materials and Methods” section P3 – L138 the sentence was written as follows: All patients underwent an internal and rheumatological visit and autoimmune and inflammatory diseases were ruled out based on clinical examinations”.

5)I also suggest that the article should be checked by a native English speaker as there are too many grammar and spelling errors.

The article was checked by a native English speaker using the MDPI Language Editing Service.

6)The author’s contribution does not state who actually performed the study. Moreover I do not see the justification for which all authors are main authors. This may be considered a form of scientific misconduct.

For research articles with few authors the Institutional Policy of Hemostasis/Hematology Unit, University of Catania, assignes the role of co-first or co-last to each author.

Round 2

Reviewer 2 Report

Dear authors,

  • line 59-61 - in the sentence ''Vitamin  K-antagonists (VKAs) have been studied retrospectively, reporting a high annual incidence of VTE (5.6--6.5)'' the incidence of VTE is not clear 5.6--6.5%?
  • line 95 - the sentence ''No patient had European Leukemia Net (ELN) disease determined to be high risk'' is ambiguous. Please clarify!
  • line 246 - in table 2 it is mentioned that 10 patients had deep vein thrombosis, but the percentage indicated is 100%. Please correct!

Author Response

Revised Manuscript Diagnostics-1676439 – Minor Revisions

Article “Impact of the Anti-Endothelial Cell Antibodies (AECAs) in Patients with Polycythemia Vera and Thrombosis”

point-by-point summary of responses

Author’s Reply to the Review Report (Reviewer 2)

1)line 59-61 – in the sentence “Vitamin K-antagonists (VKAs) have been studied retrospectively, reporting a high annual incidence of VTE (5.6--6.5)” the incidence of VTE is not clear 5.6—6.5%?

Line 59-61 - The sentence has been corrected and written as follows:

“…..reporting an annual incidence of VTE of 5.6--6.5%”.

2)line 95 – the sentence “No patient had European Leukemia Net (ELN) disease determined to be high risk” is ambigous. Please clarify!

Line 95 – The sentence has been corrected and written as follows:

“No patient had European Leukemia Net (ELN) determined high risk disease”

3)line 246 – in table 2 it is mentioned that 10 patients had deep vein thrombosis, but the percentage indicated is 100%. Please correct!

Line 246 – In table 2 the percentage has been corrected and written as follows:

“25%”